# Numerical Simulation and Experimental Investigation of the Preparation of Aluminium Alloy 2A50 Semi-Solid Billet by Electromagnetic Stirring

**DOI:** 10.3390/ma13235470

**Published:** 2020-11-30

**Authors:** Yongfei Wang, Shengdun Zhao, Yi Guo

**Affiliations:** 1School of Mechanical Engineering, Xi’an Jiaotong University, Xi’an 710049, China; wangyongfei324@mail.xjtu.edu.cn (Y.W.); sdzhao@mail.xjtu.edu.cn (S.Z.); 2State Key Laboratory of Fluid Power and Mechatronic Systems, Zhejiang University, Hangzhou 310027, China; 3School of Energy and Power Engineering, Xi’an Jiaotong University, Xi’an 710049, China

**Keywords:** magnetohydrodynamics, numerical simulation, microstructure, aluminium alloy, semi-solid

## Abstract

Electromagnetic stirring (EMS) has become one of the most important branches of the electromagnetic processing of materials. However, a deep understanding of the influence of the EMS on the thermo-fluid flow of the aluminium alloy melt, and consequently the refinement of the microstructure is still not available. This paper investigated the influence of the operating parameters of EMS on the magnetohydrodynamics, temperature field, flow field, and the vortex-shaped structure of the melt as well as the microstructure of the aluminium alloy 2A50 billet by numerical simulation and experiments. The operating parameters were categorised into three groups representing high, medium, and low levels of Lorentz forces generated by EMS. The numerical simulation matched well with the experimental result. It was found that a high level of EMS can improve the uniformity of the temperature and flow fields. The maximum speed was observed at the radius of around 25 mm under all EMS levels. Both the depth and diameter of the vortex-shaped structure generated increased with the enhancement in the EMS level. The average grain size of the edge sample of the billet was reduced by 48.3% while the average shape factor was increased by 51.0% under the medium-level EMS.

## 1. Introduction

Aluminium alloy has been widely used in various engineering applications due to its excellent physical properties such as high strength, low thermal expansion, low density, good cutting capacity, and good thermal conductivity [1,2]. As one of the well-known aluminium alloys, 2A50 is a widely used medium-strength aluminium-copper deformable alloy with high specific strength [3], high creep resistance [4], and excellent formability [5], the hardness of which is equivalent to that of duralumin after the quenching and artificial ageing heat treatment. Aluminium alloy 2A50 is often used to manufacture die forgings with complex shapes and medium strength [6], such as various aluminium alloy components in the automotive industry [7,8]. Aluminium alloy 2A50 has also been used for the production of wires as it has excellent heat conduction and corrosion resistance. However, the intergranular segregation of Cu, Mn, and Fe is a severe issue with aluminium alloy 2A50 [9]. Although the T6 heat treatment strengthening can increase the hardness of aluminium alloy 2A50, the alloy elements are severely precipitated from the inside of the grains, resulting in a decrease in mechanical properties such as elongation and tensile strength.

The electromagnetic stirring (EMS) is a contactless technique commonly applied to produce the forced convection and adjust the fluid flow during the solidification of the metal melt [10,11]. The working principle of EMS is the same as an induction motor containing a stator and a rotor [12]. The induced magnetic field is produced when the three-phase alternating current is supplied to the windings of the stator. The induction current is generated when the rotor cuts the induced magnetic field, the direction of which is perpendicular to the inducted magnetic line, and therefore generates the Lorentz force to drive the rotor. EMS is initiated when the rotor is replaced by the melt metal bulk, where the metal liquid rotates instead of the rotor. Comparing the traditional stirring, EMS has become one of the most important branches of electromagnetic processing of materials [13], which has numerous advantages for the flow control and grain refinement including the continuous production [14], easy operation [15], non-pollution [16], and non-oxidation [17]. EMS has become one of the main methods for the production of the semi-solid slurry or billets. By adjusting the operating parameters, EMS is able to remarkably improve the macroscopic quality of the metal billet [18,19].

The effects of the EMS technology on aluminium alloys have been investigated by researchers for improving the microstructure and the mechanical property of it. The grain refinement and the solidification microstructures of Al–Cu–Co with EMS were studied by Çadırlı et al. [20], which found that the increase in the magnetic strength assisted the grain refinement and uniformity of solidification microstructures. This refinement of the solidification microstructure resulted in a distinct increase in the grain boundary. Zuo et al. [21] investigated how EMS influenced the grain refinement of the aluminium alloy by studying the temperature field during the casting process. It was found that a high cooling rate was observed with the EMS technique, which improved the uniformity of the temperature field of the entire melt in the mould and consequently enhanced the heterogeneous nucleation. Mapelli et al. [22] investigated the effects of the winding current of the EMS on the pattern of flow streamlines in the liquid pool. It was concluded that the circumferential motion of the flow was changed into the radial movement by increasing the current intensity, and therefore, the velocity field of the liquid bulk can be controlled by adjusting the winding current values. Research showed that the essential reason for the refinement of the microstructure of aluminium alloy by EMS technology is the intervening of the temperature and flow fields of the metal melt by EMS [19,23]. Furthermore, the turbulent flow would be generated when EMS technology is supplied [24]. However, a deep understanding of the influence of the EMS on the thermo-fluid flow and the vortex-shaped structure of the semi-solid aluminium alloy, and consequently the refinement of the microstructure of it is still not available. 

As mentioned above, the heat and mass transfer of the slurry under the influence of EMS is vital to determine the billet geometry and solidification microstructure. Therefore, the integrated and quantitative description of the thermodynamic behavior of the slurry is necessary to provide a deep insight into the physical mechanism of EMS. However, experimental investigation on the thermos-physical information in the slurry pool during the EMS is an extremely difficult task due to the non-transparent feature of the semi-solid slurry. Thus, the simulation of EMS is necessary to provide a deep understanding of the thermos-physical characteristics under different operating conditions. Combining with the experimental results of the microstructure under these operating conditions, it can reveal how the thermodynamic behavior of the slurry influence the final microstructure of the semi-solid billet prepared by the EMS process.

Therefore, in this study, the EMS technique is proposed to prepare the aluminium alloy 2A50 semi-solid billet. The influence of the operating parameters of EMS on the magnetohydrodynamics, temperature field, flow field, and the vortex-shaped structure of the semi-solid melt as well as the microstructure of the final product is investigated by numerical simulation and experimental investigation.

## 2. Materials and Methods

### 2.1. Methodology

#### 2.1.1. Simulation Method

The EMS preparation process of the semi-solid aluminium alloy billet is simulated by using ANSOFT 15.0 and FLUENT 15.0 software (Ansys Inc., Canonsburg, PA, USA), which is able to investigate the influence of the studied operating parameters on the electromagnetic field, flow field, temperature field, and vortex-shaped structure formed on the top surface of the billet during the EMS process. The research method applied for the magnetic-current-thermal multi-physics analysis is illustrated in Figure 1. The magnetic field analysis under different winding currents and electrical frequencies is performed by ANSOFT to calculate the Lorentz force under different operating parameters. Based on the results of the Lorentz force, the stirring force levels and corresponding operating conditions are determined. After that, the geometric model in the ANSOFT is imported into FLUENT and meshed by FLUENT. The node data of the mesh in FLUENT is then export to ANSOFT using UDF (user-defined function). The Lorentz force at each node is then obtained in ANSOFT, which is then imported as the momentum source term into FLUENT through UDF. Finally, the flow field, temperature field, and the formation of the vortex-shaped structure at the top surface of the ingot during the EMS process are analysed by FLUENT. The multi-physics simulation is verified by the vortex-shaped structure of the aluminium alloy billet from the experiments.

The VOF (Volume of Fluid) model in FLUENT is used to calculate the movement of the slurry surface during the preparation of aluminium alloy semi-solid billet under different EMS force levels. The intensity of the EMS process can be judged by the shape and depth of the vortex-shaped structure generated at the slurry level. In the VOF model, the volume fraction of the pure aluminium alloy slurry is set to 1 while the volume fraction of the pure air is set to 0. The volume fraction of the two-phase mixing zone is then between 0 and 1. The inner wall of the stirring chamber is regarded as a static wall surface during the EMS simulation in FLUENT. The semi-implicit method for pressure linked equations (SIMPLE) is used in the simulation because the interface between the slurry and the inner wall is in a non-sliding shear state.

#### 2.1.2. Model Assumptions

The EMS is a complex high-temperature time-varying process, which is a three-field coupling process of the electromagnetic field, temperature field and flow field. This three-field coupling would affect the heat transfer, mass transfer, and the viscosity during the solidification process, which further influences the phase change and the release of the latent heat from crystallization. The key assumptions for the numerical modelling of the EMS are as follows.

(a)The influence of displacement current and metal paste movement on the electromagnetic field is not considered.(b)The influence of the cooling water pipe, the insulation layer, and the corundum pipe on the magnetic field is negligible.(c)The Joule heat generated by the electromagnetic field in the slurry is ignored due to the low stirring frequency and short stirring time.(d)The metal slurry is regarded as an incompressible single-phase non-Newtonian fluid.(e)The average EMS force during one cycle is used to couple with other field variables instead of the instantaneous EMS force.

#### 2.1.3. Simulation Model

The geometric model of the electromagnetic stirrer is shown in Figure 2a, which is mainly composed of the three-phase two-pole stirrer, the cooling water pipe, the insulation layer, the resistive heater, the corundum tube, the stainless steel crucible, and the thermocouple. The three-phase two-pole stirrer is the core component of the electromagnetic stirrer containing a yoke and the coils as shown in Figure 2b. The coils adopt 36-slot double-stacked winding, with 28 wires per slot. The rated power of the three-phase two-pole stirrer is 30 kW while the rated speed is 2950 r/m. The dimensions of the key parts of the electromagnetic stirrer and the prepared aluminium alloy blank are summarised in Table 1.

The three-dimensional numerical simulation model of the three-phase two-pole electromagnetic stirrer is shown in Figure 3. Considering the low magnetic permeability in the insulation layer, the corundum tube, and the stainless steel crucible, the effect of which on the internal magnetic field of the aluminium alloy slurry is negligible, the height of these components are therefore built as the same height as the alloy blank. The thermophysical parameters of materials required in the numerical simulation of the EMS for the aluminium alloy slurry are summarised in Table 2.

#### 2.1.4. Governing Equations

The analysis of the electromagnetic field of the stirrer in the numerical simulation is based on the Maxwell equations, which specifically include Equations (1)–(3). The electromagnetic field is regarded as a steady-state harmonic electromagnetic field as the influence of the metal melt flow field on the electromagnetic field is not considered in this study. The instantaneous EMS force cannot be directly derived as the EMS force changes periodically in a harmonic analysis by the ANSOFT software. Therefore, the average EMS force in one period is calculated and analysed by Equation (4). It is worthy to note that all the EMS force in this study refers to the average force in one period.
(1)$$\;\nabla \times \vec{H}=\vec{J},$$
(2)$$\nabla \times \vec{E}=-\frac{\partial \vec{B}}{\partial t},$$
(3)$$\nabla \times \vec{B}=0,$$
(4)$$\vec{F}=\vec{J}\times \vec{B},$$
where $\vec{H}$ is the magnetic intensity, $\vec{J}$ is the induced current density, $\vec{E}$ is the electric field strength, $\;\vec{B}$ is the magnetic induction intensity, and $\vec{F}$ is the EMS force.

The thermo-flow of the metal melt during the EMS can be analysed through the continuity equation, Navier–Stokes equation, and energy conservation equations, which are shown in Equations (5)–(7), respectively.
(5)$$\nabla \times \vec{u}=0,$$
(6)$$\rho \left(\frac{\partial \vec{u}}{\partial t}+\vec{u}\cdot \nabla \vec{u}\right)=-\nabla p+\mu \nabla^{2}\vec{u}+\vec{F},$$
(7)$$\rho c_{p}\left(\frac{\partial T}{\partial t}+\vec{u}g\nabla T\right)=\nabla g\left(\lambda \nabla T\right)+\rho L\frac{\partial f_{s}}{\partial t},$$
where ρ is the density, $c_{p}$ is the specific heat capacity, *t* is the time, *P* is the pressure, *T* is the temperature, $\vec{u}$ is the velocity, λ is the heat conductivity coefficient, μ is the dynamic viscosity, *L* is the latent of the melt, and $f_{s}$ is the solid fraction.

The formation of vortices on the surface of the molten metal and the variation of the slurry level under the EMS force is calculated based on the VOF (Volume of Fluid) equation as shown by Equation (8).
(8)$$\frac{\partial f}{\partial t}+\vec{u}\cdot \nabla f=0,$$
where *f* is the phase volume fraction.

Considering the influence of temperature drop and shear rate on the apparent viscosity of the slurry during the EMS process, the Power Law Cut-Off (PLCO) viscosity model as shown in Equation (9) is adopted in this study for calculating the viscosity of the slurry.
(9)$$\{\begin{array}{c} \mu \left(\dot{\gamma },T\right)=\mu_{0}\left(T\right){\dot{\gamma }}_{0}^{n\left(T\right)}\;\;\;\;\;\;\mathrm{if}\;\dot{\gamma }\leq {\dot{\gamma }}_{0}\\ \mu \left(\dot{\gamma },T\right)=\mu_{0}\left(T\right){\dot{\gamma }}^{n\left(T\right)}\;\;\;\;\;\;\;\mathrm{if}\;\dot{\gamma }>{\dot{\gamma }}_{0} \end{array},$$
where $\mu_{0}$ is the temperature-dependent reference viscosity, $\dot{\gamma }$ is the shear rate, ${\dot{\gamma }}_{0}$ is the critical shear rate, and n is the shear thinning index.

### 2.2. Experimental Design

#### 2.2.1. Experimental Setup and Materials

An EMS test system was set up for the validation of the simulations and the microstructure investigation, which was mainly composed of five parts: an intermediate frequency induction melting furnace, a three-phase two-pole electromagnetic stirrer, a water cooling system, a crucible preheating system, and a control system. The intermediate frequency induction furnace was used to quickly heat the studied alloy due to the advantages of high energy efficiency, fast heating speed, and a high degree of automation. The support frame was designed for the convenient loading and unloading of the melting crucible. The cooling water was used to eliminate the overheating of the induction coil. The infrared thermometer was applied to monitor the slurry temperature in real-time, according to which the power of the intermediate frequency induction furnace was appropriately adjusted for heating the studied alloy at the designed temperature. More information about the intermediate frequency induction furnace can be found in Table 3. 

The structure of the electromagnetic stirrer is shown in Figure 4. The Y200L1-2 three-phase two-pole motor stator (Xi’an Sanrui Electric Furnace Co. LTD., Xi’an, China) was applied as the electromagnetic stirrer as it can produce a relatively uniform electromagnetic field, which had the rated power of 30 kW and the rated speed of 2950 r/min. The external and internal diameters of the motor stator were 327 and 182 mm, respectively while the core length of it was 180 mm. The motor stator adopted the double-stacked 36-slot winding with 28 wires per slot. The E200-045T3 inverter (Xi’an Sanrui Electric Furnace Co. LTD., Xi’an, China) V-f control mode was adopted to realise the independent control of the stirring frequency and stirring current during the EMS process. The inverter used in this study had the rated input voltage of 380 V, the rated current of 90 A, the rated power of 45 kW, and the frequency adjustment range of 0.05 to 600 Hz.

The cooling water pipes and the insulation layer were designed in the electromagnetic stirrer since the temperature of the slurry pouring into the crucible was higher than 600 °C and the working temperature of the motor stator cannot exceed 70 °C. The molybdenum wire was wound on the outside of the stirring chamber with an insulation layer to avoid the cooling of the aluminium alloy slurry due to the low temperature of the stirring chamber. The preheating temperature of the stirring chamber was controlled to not exceed 500 °C to prevent excessive oxidation of the molybdenum wire through adjusting the electricity in the molybdenum wire by proportion integration differentiation (PID) control with the maximum power of 3.6 kW, rated voltage of 60 V, and rated current of 60 A. The tapered graphite sprue sleeve and the graphite drain tube were adopted to guide the flow of the slurry pouring into the stirring chamber due to the non-adhesive characteristics of graphite and metal, which was helpful to obtain a stable pouring process and eliminate the pouring entrainment.

Commercial aluminium alloy 2A50 extruded bar supplied by Aluminum Corporation of China (Beijing, China) was selected as the test material in this study. Its composition is summarised in Table 4, which shows that the main alloying element is copper. The solidus and liquidus temperatures are 521 and 615 °C, respectively. This means that the semi-solid temperature range is as wide as up to 94 °C, which is suitable for semi-solid forming. 

#### 2.2.2. Experimental Procedure

The experimental procedure of preparing a semi-solid slurry of aluminium alloy 2A50 by the EMS method is presented in Figure 5. The extruded bar material of aluminium alloy 2A50 was first cut in sections and placed in the melting crucible. The intermediate frequency induction heating furnace was then turned on to quickly heat the aluminium bar to the predetermined melting temperature, which needs to be maintained for 30–45 min to ensure the raw material was completely melted and the temperature was uniform in the melt. At the same time, the molybdenum resistance heater and the water cooling system were turned on to preheat the stirring chamber and cool the motor stator, respectively. When the temperature of the stirring chamber reached the pre-set value, the aluminium alloy melt was quickly poured into the stirring chamber through the funnel-shaped graphite cone sleeve. The electromagnetic stirrer then started to work at the pre-set winding current and electrical frequency until the melt was completely solidified. The semi-solid slurry after the EMS treatment should be quickly quenched and cooled into billets in order to prevent the crystal grains from being too coarse.

In order to observe and analyse the microstructure of the semi-solid slurry, samples were taken at the centre and edge of the semi-solid aluminium alloy bar prepared by EMS as shown in Figure 6. Samples were cut as 8–10 mm cubes and then inlaid as cylindrical components with a diameter of 30 mm and a height of 25 mm. After that, they were processed with the rough grinding, fine grinding, polishing, and finally chemical etching with 0.5% hydrofluoric acid for 12–20 s. The chemical etching had been finished, the microstructure of samples was observed by the Olympus optical microscope. The area and circumference of the crystal grains in the microstructure were then measured by the Image-Pro Plus image analysis software. In order to accurately analyse the average diameter and shape factor of the crystal grains of each sample, at least 3 fields of view of each sample were taken for observation. In each field of view, the area and perimeter of the crystal grains were measures at more than 5 different positions. The average size of grains (D) and shape factor (F) were calculated by Equations (10) and (11) [25,26,27] based on the number, areas, and perimeters of all solid grains in the sample obtained by using the Image-Pro Plus 6.0 software. The average diameter is the average value of the equivalent diameter of all grains. The equivalent diameter is the diameter of a circle, of which the area is equal to the area of the grain. The error band of the average size and shape factor were calculated by Equations (12) and (13), respectively.
(10)$$\bar{D}=\frac{\sum_{1}^{N}\sqrt{4A/\pi }}{N},$$
(11)$$\bar{F}=\frac{\sum_{1}^{N}4\pi A/P^{2}}{N},$$
(12)$$S_{D}=\sqrt{\frac{1}{N-1}\sum_{1}^{N}{\left(D_{N}-\bar{D}\right)}^{2}},$$
(13)$$S_{F}=\sqrt{\frac{1}{N-1}\sum_{1}^{N}{\left(F_{N}-\bar{F}\right)}^{2}},$$
where, *A* is the area of the crystal grain, *D* is the diameter of the crystal grain, *F* is the shape factor of the crystal grain, $\bar{D}$ is the average diameter of grains, *P* is the perimeter of the crystal grain, $\bar{F}$ is the average shape factor of grains, *N* is the number of grains, $S_{D}$ is the error band of the average diameter, and $S_{F}$ is the error band of the average shape factor.

## 3. Results and Discussion

### 3.1. Verification of the Numerical Model

The comparison of the vortex-shaped structure for one sample between the experimental and the simulation calculated results with the winding current of 30 A and the electrical frequency of 30 Hz is presented in Figure 7. It can be seen in Figure 7a that the vortex-shaped structure was observed obviously at the top surface of the overview of the experimental result. The shrinkage porosity and cavity were observed occupied between the air interface line and the solid interface line of the vortex-shaped structure in the experimental vertical section. In order to clearly compare the shape characteristics, three pictures are presented to show the difference in the shape line, maximum diameter, and the depth of the vortex-shaped structure. Comparing the vertical sections of the vortex-shaped structures, the profile of the vortex-shaped structure from the simulation matched well with the experimental result. It was found from Figure 7b that the diameter of the experimental vortex-shaped structure top was around 56 mm while the simulation calculated one was about 51 mm, representing an error of 8.9%. The depth of the vortex-shaped structures from the experimental and simulation results was found as approximately 88 and 92 mm, respectively, indicating a deviation of 4.5%.

### 3.2. Results of the Magnetohydrodynamics Analysis

The electromagnetic intensity, induced current density, and Lorentz force distributions at the excitation winding current of 50 A, electrical frequency of 50 Hz, and the phase angle of 0° are shown in Figure 8a–c, respectively. The operating conditions of 50 A and 50 Hz are chosen as an example to show the magnetohydrodynamics results as they are the maximum operating conditions in this study. It can be seen from Figure 8a that the maximum and the minimum values of the electromagnetic intensity are all observed at the outside surface of the cylinder billet. The electromagnetic intensity at the top and bottom surfaces, as well as the outside cylindrical surface, were all higher than the interior electromagnetic intensity, which represents the surface and end effects. Figure 8b shows that the vectors of the induced current form closed loops at both the outside surface and interior region of the billet. The distribution pattern of the induced current density is similar to the electromagnetic intensity, which also shows the surface and end effects. It can be found in Figure 8c that the Lorentz force distribution presents more severe surface and end effects. The minimum value of the Lorentz force occupied the majority of the interior region of the billet while the maximum was observed at the outside surface of the slurry. 

The effect of the winding current on the radial and axial distribution of the electromagnetic intensity along the symmetric line is represented in Figure 9. It was found that the increase in the winding current resulted in the enhancement in the electromagnetic intensity. This effect was weakened at the middle inner part of the slurry when the electrical frequency increased comparing Figure 9a–c or Figure 9d–f. The increase in the electrical frequency also decreased the electromagnetic intensity at the same location through the whole slurry field. The surface effect was not obvious when the electrical frequency is 10 Hz as shown in Figure 9a, which was intensified by the increase in either the winding current or the electrical frequency comparing Figure 9a–c. The end effect was also reinforced by the augment in the winding current or the electrical frequency represented by Figure 9d–f.

Figure 10 shows the simulation results of the Lorentz force along the symmetric line at different winding currents and electrical frequencies. It was found that there was no obvious effect on the Lorentz force at the central position of the slurry when the winding current increased while it significantly increased the Lorentz force at the outer and end surfaces. This means that the surface and end effects were strengthened by the increase in the winding current. As shown in Figure 10a–c, the increase in the electrical frequency improved the uniformity of radial force at the central part of the cylinder specimen while almost no influence was observed on the radial force at the radius of 30 mm. Comparing Figure 10d–f, the increase in the electrical frequency aggravated the end effect although the uniformity of the Lorentz force was significantly improved in the central part of the slurry along the axis. It was also found from Figure 10d–f that the end effect was also strengthened when the winding current increased at all electrical frequency cases.

The results of the average Lorenz force along the radius and the axis at different winding currents and electrical frequencies are shown in Figure 11. It was found that both the average Lorenz force and its standard deviation increased with the increase in the winding current. The effect of the electrical frequency on the average Lorenz force along both the radius and axis was not obvious when the winding current was as low as 10 Hz. However, when the current increased, the difference of average Lorentz force at different electrical frequencies was enlarged, where the larger the electrical frequency was supplied the lower the average Lorentz force was observed. At a certain winding current, although the standard deviation first enlarged with the increase in the electrical force and then decreased with a further increase, the difference in the average Lorentz forces was not obvious. This means that the influence of the winding current on the standard deviation of the Lorentz force was higher than the electrical frequency, which aggravated the non-uniformity of the Lorentz force distribution along both the radius and axis.

Results of average Lorentz force along the radius at different winding currents and electrical frequencies are summarised in Figure 12 for analysis as the average Lorentz force along the radius was higher compared to that along the axis as shown in Figure 11. The results were categorised into three groups representing high, medium, and low levels of Lorentz forces, which is presented in Figure 12. Three cases with typical operating values of electrical frequency and winding current were selected to represent the high, medium, and low levels of Lorentz forces produced due to the relatively low standard deviation. The low-level force case was determined as the electrical frequency of 10 Hz and winding current of 10 A while the medium-level force case was determined as the electrical frequency of 30 Hz and winding current of 30 A. The electrical frequency of 50 Hz and the winding current of 50 A were selected as the operating conditions of the high-level force case. The temperature distribution analysis and the flow field analysis were then conducted at the aforementioned three levels.

### 3.3. Effects of the Operating Conditions on the Temperature Field

Figure 13 shows the variation of temperature field with time for the sectional view at the half-height of the slurry under the high-level EMS. Circle patterns were observed with the variation of time, where the temperature of the outer circles was lower than the central region. The temperature at the central region decreased to 828.2 K while that at the outer circle was cooled down to 834.2 K at the time of 19.37 s.

The results of the temperature field for the sectional view at the half-height of the slurry under different operating conditions at the time of 14.56 s are shown in Figure 14. With the enhancement in the EMS level, both the temperature at the central region and the temperature difference between the central and the edge of the slurry decreased due to the heat transfer through the forced convection. It was also found that the number of temperature circles decreased with the intensification in the EMS level, which means that the uniformity of the temperature in the central region of the slurry was also improved with the strengthening in the Lorentz force. 

The temperature variation with time at the centre of the slurry and the temperature variation along the radius at 20 s for the sectional view at half-height are shown in Figure 15a,b, respectively. It can be found in Figure 15a that the decrease rate of temperature increased with the enlargement in the EMS level. It required 48 s for the centre of the slurry to be cooled down to 550 K while only 30 s was needed when the high level of Lorentz force was supplied by the EMS. As shown in Figure 15b, the temperature of the slurry at the radius of 0 mm decreased from 587.0 K in the case without the EMS to 573.5 K in the high-level EMS case. At the edge of the slurry (i.e. the radius of 30 mm), the temperature also decreased with the strengthening in the Lorentz force, which was found as 574.0 and 569.0 in the cases without electromagnetic force and high-level EMS, respectively. In the major region of the central part, from the radius of 0 to the radius of 25 mm, the slope of the temperature curve decreased with the increase in the EMS level. Under the medium and high levels of EMS, the temperature maintained stable from the radius of 0 to the radius of 25 mm as represented in Figure 15b.

### 3.4. Effects of the Operating Conditions on the Flow Field

The variation of the velocity field for the cross-section of the slurry at half-height during the acceleration stage of the EMS is shown in Figure 16. It can be found that at the beginning of the acceleration stage, there was no obvious increase in the speed of the central region due to the small stirring force. Therefore, the velocity growth area of the slurry cross-section was mainly distributed in an annular area ranged from about 1/2 to 2/3 of the radius. The speed of this annular region increased while the coverage area of it enlarged with the EMS continuing. The area of the annular area reached the maximum value when it was close to the end of the acceleration stage. The high-speed rotating annular area can drive the slurry in the central region continuously accelerate the rotation, which consequently caused the increase in the speed of the central area of the slurry. At the end of the acceleration stage, the maximum speed occupied almost 3/4 of the total cross-section area.

Figure 17 shows the flow fields of the cross-section at the half-height of the slurry at the end of the acceleration procedure under different levels of EMS. It was found that the velocity distribution pattern was almost the same at all levels of EMS, where the maximum velocity was observed at the middle circles with low velocities at the central and outer circles. The maximum velocity of the slurry was found as 0.27 m/s under the low level of EMS while that increased up to 1.6 and 5.0 m/s with the medium and high levels of EMS, respectively. It was also found that the high-velocity area gradually increased with the enlargement in the EMS level. The increment in the velocity is helpful to fragment the dendrites and consequently, the fragments are able to flow away by the bulk liquid, which acts as nucleation sites for the refined grains.

The velocity variation with time for two specific locations and the velocity variation along the radius under different levels of EMS are shown in Figure 18a,b, respectively. The chosen two positions labelled as P1 and P2 are located at the half-height of the slurry with a radius of 0 and 25 mm, respectively. It can be found in Figure 18a that the velocity at the studied positions first increased with time and then decreased to 0 at the time of 35 s. This was because, at the beginning of the stirring, the viscosity of the slurry was relatively low due to the high temperature. With the increase in the velocity, the forced convection was obtained which cooled down the temperature of the slurry and consequently increased the viscosity. As a result, the velocity finally decreased to 0 with time. It was also found in Figure 18a that the velocity at Position P1 was around 0 all the time under the low-level EMS while it can be increased to 1 m/s at the time of 5 s under the high level of the stirring force. Under the low level of EMS, the maximum velocity of Position P2 was lower than 0.5 m/s while it was closed to 2 m/s with the medium level of the stirring force. When the slurry was supplied with the high-level EMS, the maximum velocity of Position P2 was observed over 6 m/s as shown in Figure 18a. The velocity difference between Positions P1 and P2 increased by the enhancement in the EMS level, which would result in the vortex-shaped structure at the top liquid surface. 

Figure 18b shows the velocity variation along the radius at the time of 5 s and Z = 90 mm under different levels of EMS. It was found that the maximum speed appeared at the radius of around 25 mm under all cases of EMS level although the value of the maximum velocity increased with the enhancement of EMS, which means that EMS had almost no influence on the distribution of the velocity along the radius. Under the low level of EMS, the velocity in the whole cross-section was observed significantly low with the maximum value around 0.3 m/s. When the slurry was operating under the medium level of the stirring force, the maximum velocity increased up to 1.6 m/s. When the high level of Lorentz force was generated by the EMS, the velocity of the most region of the cross-section was higher than 1.1 m/s with the maximum velocity reaching 6 m/s. 

### 3.5. Effects of the Operating Conditions on the Vortex-Shaped Structure 

The variation of the vortex-shaped structure with time under the high level of EMS is presented in Figure 19 to present the generation procedure of the vortex-shaped structure with EMS applying. It can be found that with the EMS supplied to the slurry, the liquid level fluctuation can be observed at the time of 1.25 s. At the time of 2.50 s, the level of the slurry edge rose with a shadow swirl generated in the centre. Both the diameter and the depth of the vortex-shaped structure increased with time. The diameter of the vortex-shaped structure almost occupied half size of the total diameter of the slurry at the time of 3.75 s while the liquid level of the slurry edge rose by 42 mm as shown in Figure 19. It was also found that at the time of 5.00 s, the liquid level of the slurry edge increased by 75 mm while the depth of the vortex-shaped structure occupied half-height of the slurry. As aforementioned, four stages can be observed during the vortex-shaped structure generation procedure under EMS, which is the fluctuation of the liquid level, the initiation of the vortex-shaped structure, the intensification of the vortex-shaped structure, and the final formation of the vortex-shaped structure.

The vortex-shaped structure generated under different levels of EMS at the time of 5.00 s is shown in Figure 20. It was found that under the low level of EMS, it only reached the initiation of the vortex-shaped structure and then finished the solidification process. The vortex-shaped structure generated under the medium level of EMS was intensified compared to the low-level EMS operating condition. Comparing Figure 20b,c, the vortex-shaped structure had a thicker slurry edge, a relative shadow depth, and a relatively small radius under the medium level of EMS. A significant rise of the vortex-shaped structure edge was observed with the high-level stirring force, which would generate the backflow from the vortex-shaped structure edge to the central region and consequently result in the entrainment of the gas into the slurry. This gas entrainment would further cause the porosity in the product and degrade the product quality.

### 3.6. Effects of the Operating Conditions on the Microstructure 

The microstructure of the billet prepared without EMS and that with the medium-level EMS were observed by the optical microscope (NIKON ECLIPSE LV 150N, Nikon, Tokyo, Japan) for the comparison. The result of the microstructure comparison is shown in Figure 21. It can be seen from Figure 21a that the microstructure of the alloy is mainly composed of coarse dendrites, part of which has the arm even longer than 200 μm. A large number of dendrites were intertwined with each other with the eutectic structure inside, which is the typical as-cast microstructure. The microstructure of the central and edge samples (see Figure 6) of the billet prepared by the medium-level EMS are presented in Figure 21b,c, respectively. Solid grains with the average grain size significantly smaller than 100 μm were observed, which were evenly distributed in the eutectic phase. Moreover, some nearly spherical grains were observed bonding to each other in the microstructure of the central sample shown in Figure 21b compared to the edge sample shown in Figure 21b, which can be attributed to the weak effect of the EMS at the central slurry. 

The average grain size and shape factor of the microstructure for the 2A50 aluminium alloy prepared without EMS and with the medium-level EMS are summarised in Figure 22. The average grain size of the billet was 140.7 μm while the shape factor was 0.49 when the EMS was not applied. Under the medium level of EMS, the average grain size of the central and edge samples were obtained as 78.40 and 72.80 μm, respectively. The shape factor of the central sample was found as 0.72 while that of the edge sample was observed as 0.74 with the medium-level EMS. By applying the EMS, the average grain size was reduced by 44.3% and 48.3% for the central and edge samples of the billet, respectively, while the average shape factor was increased by 46.9% and 51.0%, respectively. This means that EMS can significantly improve the microstructure of the 2A50 aluminium alloy and the distribution of the eutectic phase.

The achievement of fine and spherical semi-solid microstructures through the EMS process can be explained as follows. When the slurry was poured into the stirring chamber, the edge and bottom of the slurry directly contacted the wall of the stirring chamber, which rapidly caused the supercooling of the slurry at these contacting positions. Consequently, a large number of crystal nuclei are formed on the inner wall and bottom surface of the stirring chamber. Under the interference of the electromagnetic stirring, the slurry repeatedly washed the inner wall and bottom face due to the high-speed rotating. This washing movement of slurry forced the crystal nuclei generated to fall off and enter back to the slurry. Meanwhile, the slurry at the core space of the stirring drum was forced to the inner wall of the stirring chamber. This forced flow of slurry helped to form a uniform distribution of crystal nuclei, which can effectively inhibit the evolution of crystal nuclei to dendrites. Furthermore, the forced convection movement of the slurry can increase the local shear rate inside the stirring drum, which caused the frequent collision, rub, and shearing of grains and consequently formed the fine spherical grains. 

## 4. Conclusions

In this work, the effects of the operating parameters of EMS on the magnetohydrodynamics, temperature field, flow field, the vortex-shaped structure formation of the melt, and the microstructure of the aluminium alloy 2A50 billet are investigated by numerical simulation and experiments. The simulation is validated with the experimental result. The main conclusions can be drawn as follows.

The operating parameters were categorised into three groups representing high, medium, and low levels of Lorentz forces generated by EMS according to the magnetohydrodynamics analysis. The low, medium, and high levels of the winding current were found as 10, 30, and 50 A, respectively, while that of the electrical frequency were selected as 10, 30, and 50 Hz, respectively. 

Circle patterns were observed in the temperature field. The uniformity of the temperature field was improved with the increase in the EMS level while the cooling time of the slurry decreased. The cooling time decreased to 30 s from 615 to 550 K under the high-level EMS while it needed 50 s in the case without EMS. 

The uniformity of the flow field was improved by EMS with almost 3/4 of the cross-section area was occupied by the maximum speed at the end of the acceleration stage. The maximum speed was observed located at the radius of around 25 mm under all EMS levels although the value of the maximum velocity increased with the enhancement of EMS.

Four stages can be observed during the vortex-shaped structure generation procedure under the high level of EMS, which is the fluctuation of the liquid level, the initiation of the vortex-shaped structure, the intensification of the vortex-shaped structure, the final formation of the vortex-shaped structure. The vortex-shaped structure generated was intensified by the enhancement of EMS.

The dendrites were completely fragmented and the formation of small independent spherical crystal grains was obtained by applying the medium-level EMS. The average grain size was reduced by 44.3% and 48.3% for the central and edge samples of the billet, respectively, while the average shape factor was increased by 46.9% and 51.0%, respectively, under the medium-level EMS.

## Figures and Tables

**Figure 1 materials-13-05470-f001:**
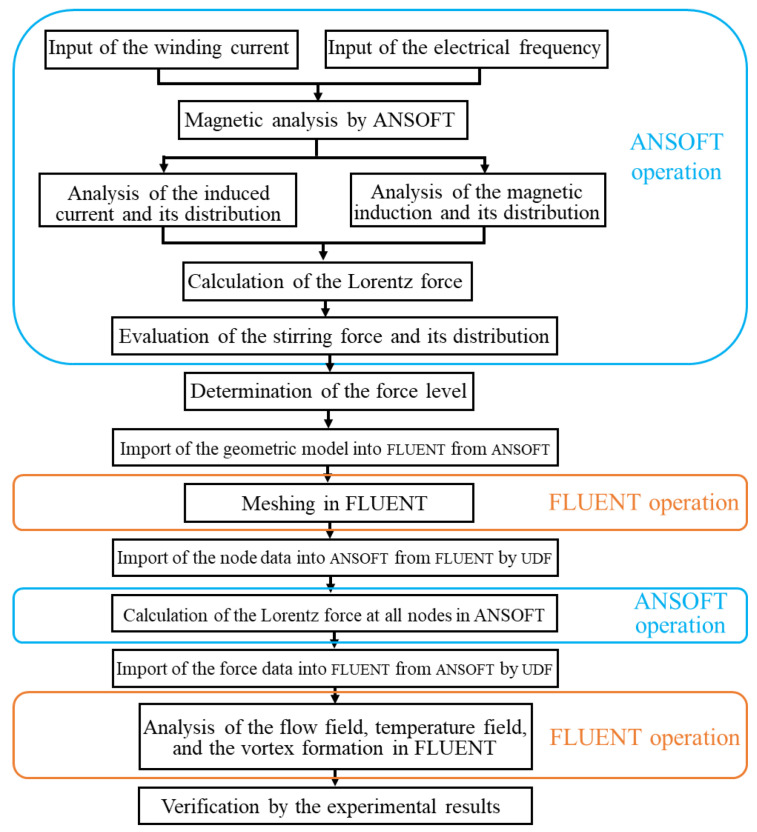
Research approach for magnetic-current-thermal multi-physics analysis.

**Figure 2 materials-13-05470-f002:**
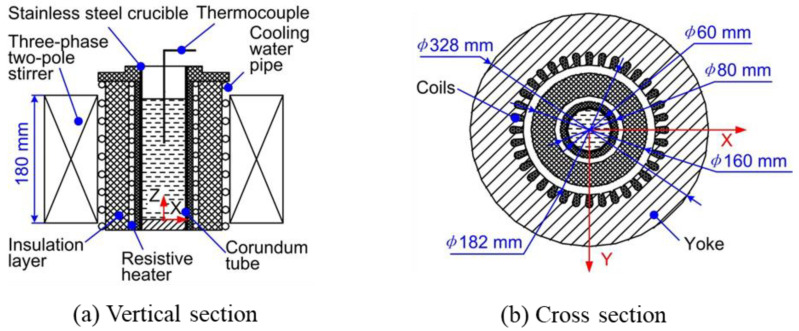
Geometric model of the electromagnetic stirrer.

**Figure 3 materials-13-05470-f003:**
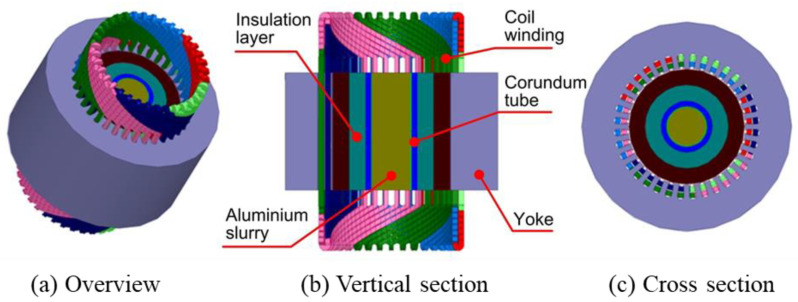
Numerical simulation model of the electromagnetic stirrer.

**Figure 4 materials-13-05470-f004:**
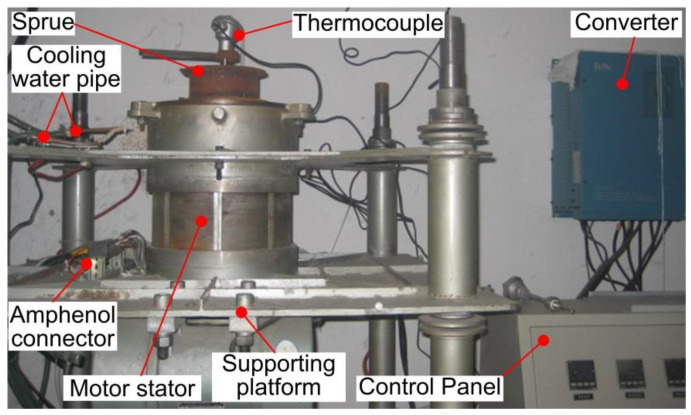
Experimental test rig of the electromagnetic stirrer.

**Figure 5 materials-13-05470-f005:**
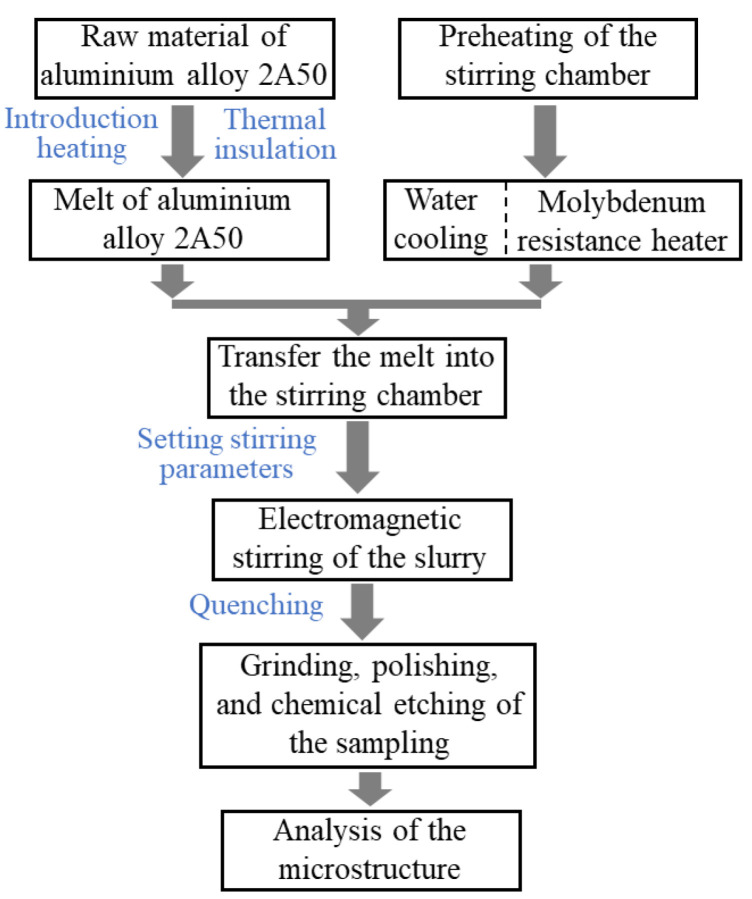
Structure schematic of the electromagnetic stirrer.

**Figure 6 materials-13-05470-f006:**
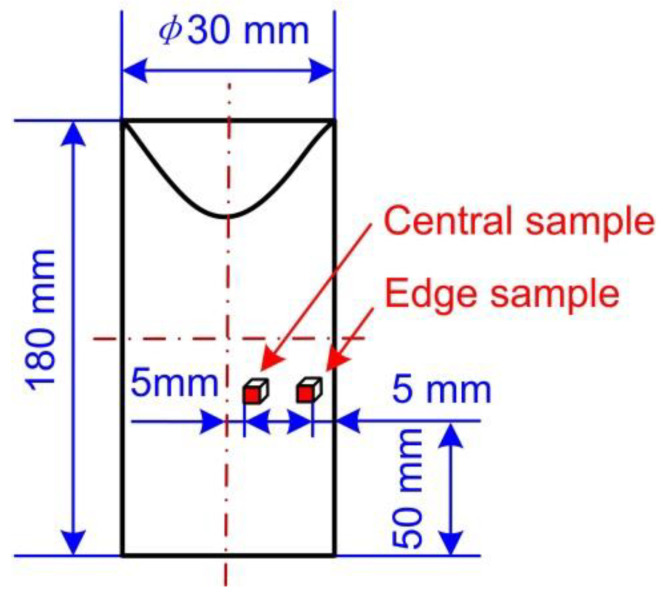
Schematic of the sampling positions.

**Figure 7 materials-13-05470-f007:**
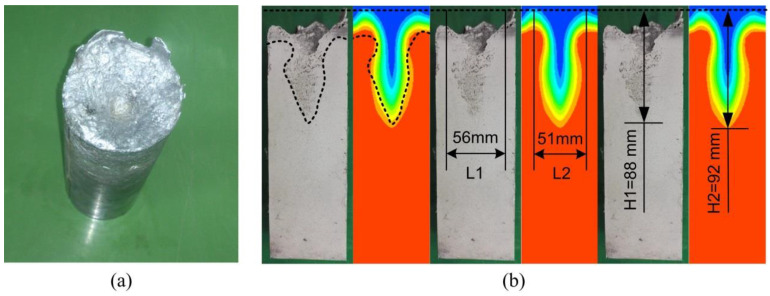
(**a**) Experimental overview and (**b**) the comparison of the vertical section between the experimental and simulation vortices.

**Figure 8 materials-13-05470-f008:**
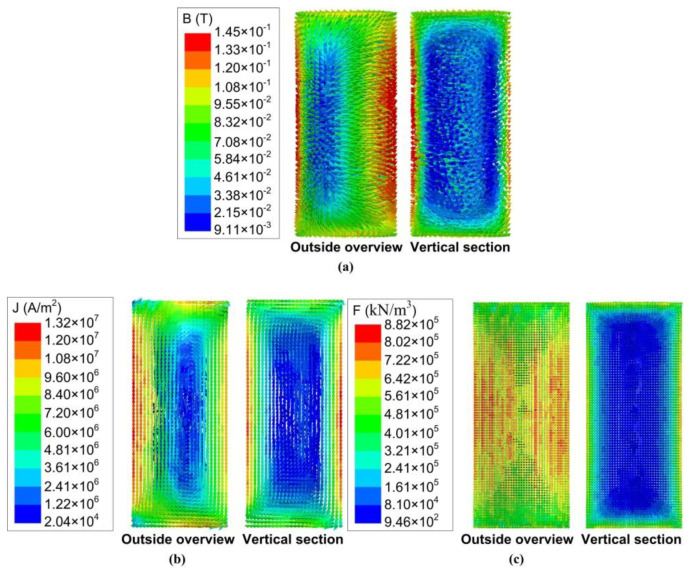
Results of (**a**) electromagnetic intensity, (**b**) induced current density, and (**c**) Lorentz force distributions—winding current: 50 A; electrical frequency: 50 Hz; phase angle: 0°.

**Figure 9 materials-13-05470-f009:**
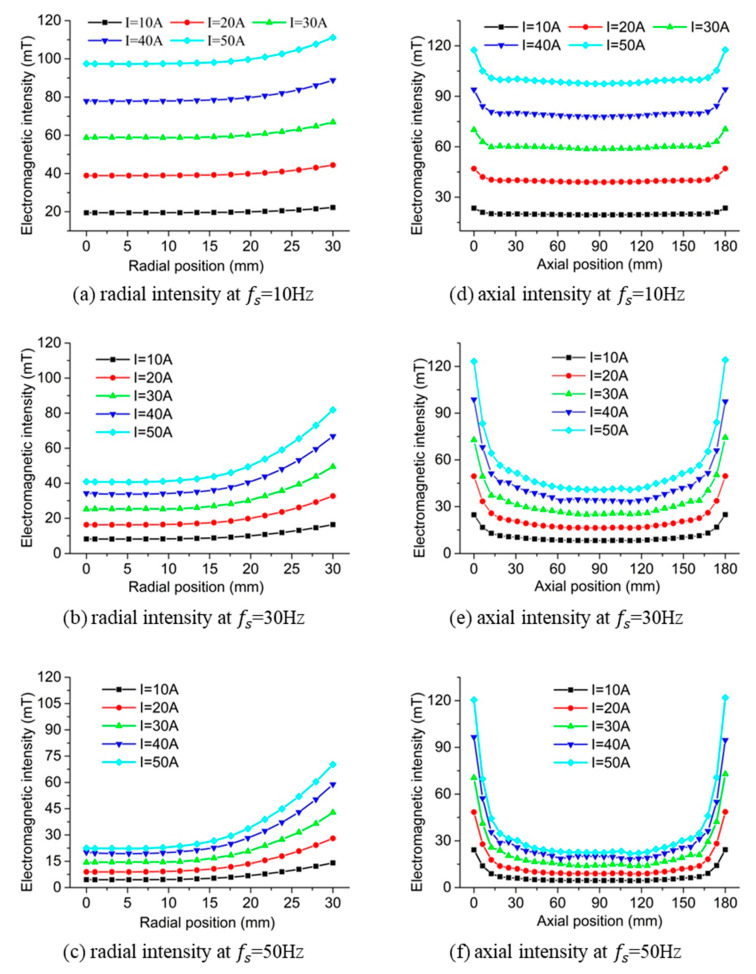
Results of the electromagnetic intensity at different winding currents and electrical frequencies.

**Figure 10 materials-13-05470-f010:**
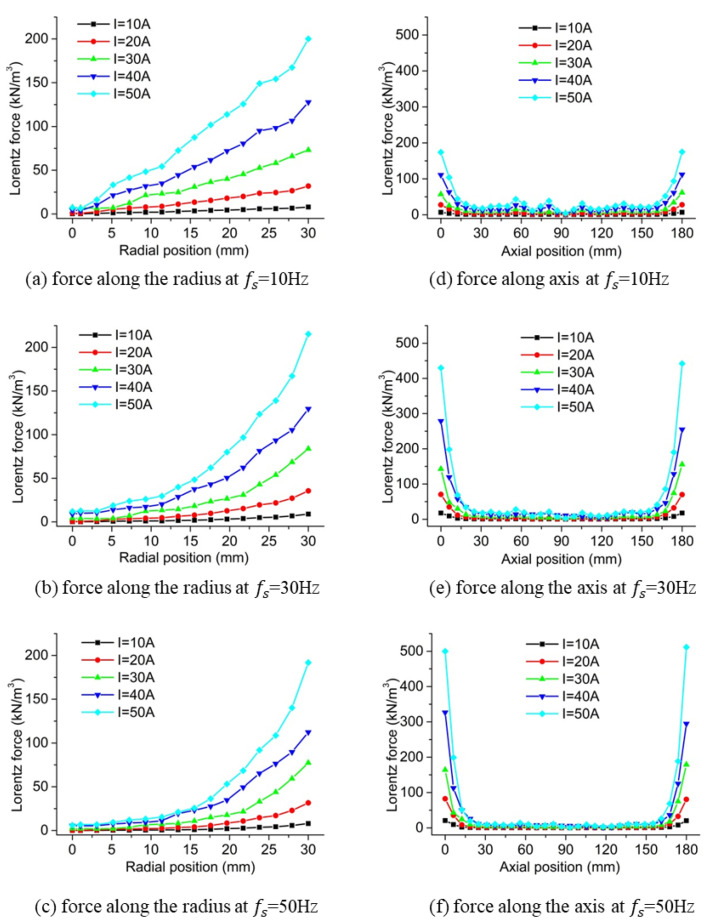
Results of the Lorentz force at different winding currents and electrical frequencies.

**Figure 11 materials-13-05470-f011:**
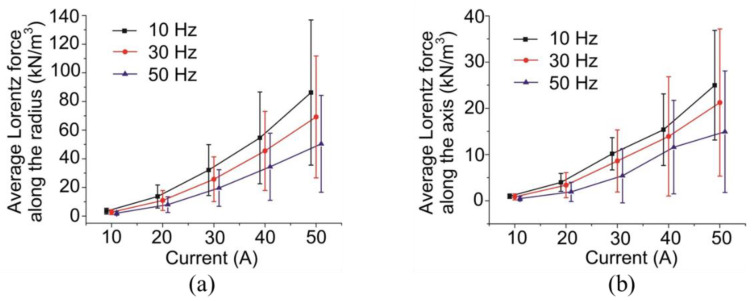
Average Lorentz force along the (**a**) radius and (**b**) axis at different winding currents and electrical frequencies. Error bars indicate the standard deviation of the average Lorentz force.

**Figure 12 materials-13-05470-f012:**
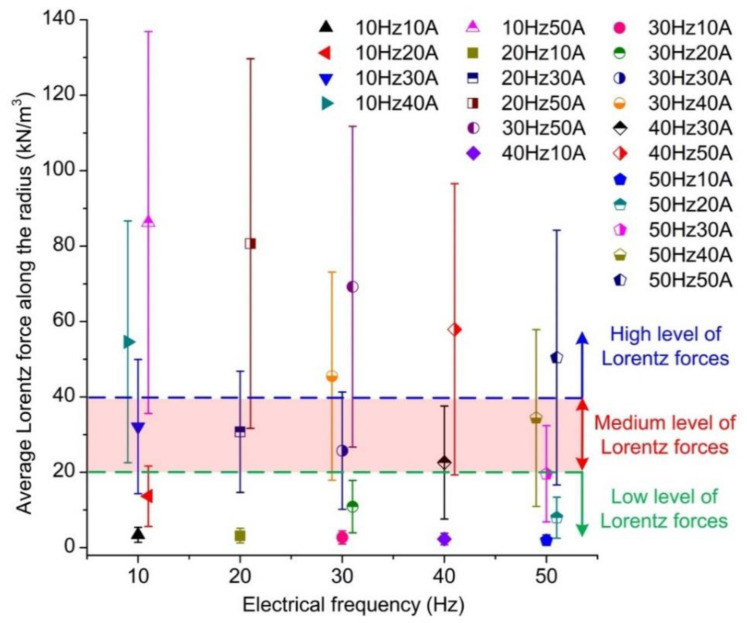
Summarisation of the average Lorentz force along the radius at different operating conditions.

**Figure 13 materials-13-05470-f013:**
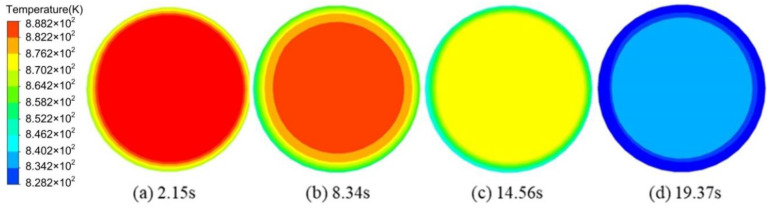
Variation of the temperature field with time under the high-level electromagnetic stirring (EMS).

**Figure 14 materials-13-05470-f014:**
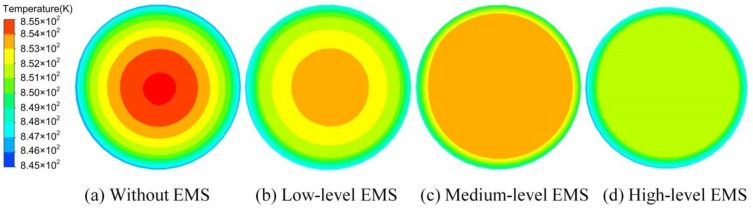
Results of the temperature field under different operating conditions (time: 14.56 s).

**Figure 15 materials-13-05470-f015:**
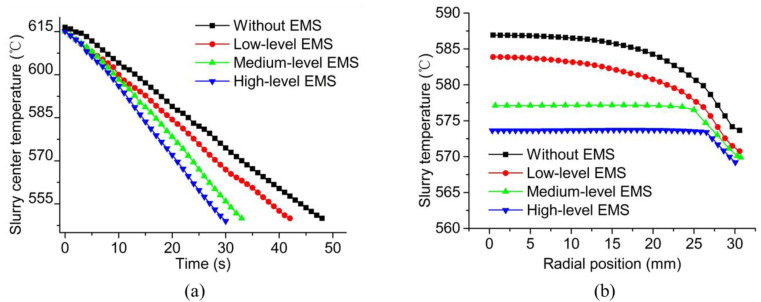
Temperature variation (**a**) with time and (**b**) along the radius at 20 s for the sectional view at half-height.

**Figure 16 materials-13-05470-f016:**
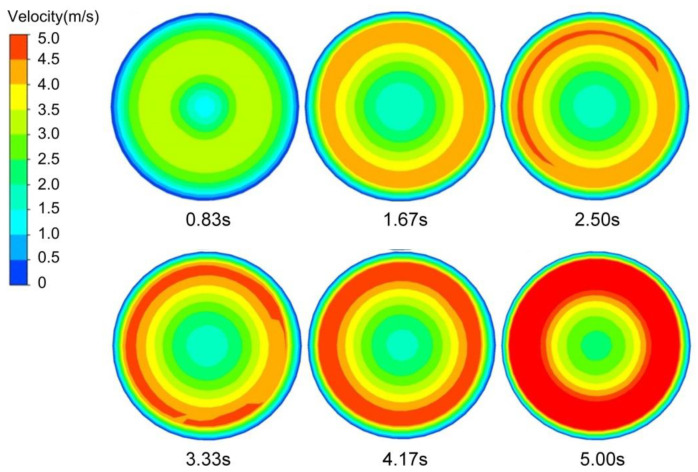
Variation of the flow field with time during the acceleration stage under the high-level EMS for the sectional view at half-height.

**Figure 17 materials-13-05470-f017:**
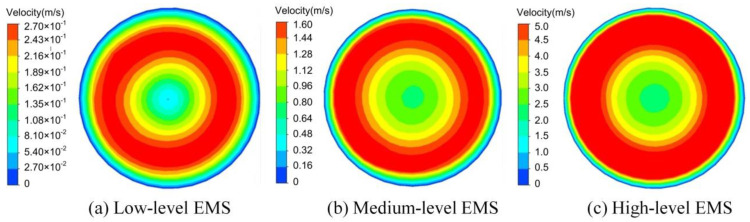
Effects of the operating conditions on the velocity and velocity distribution.

**Figure 18 materials-13-05470-f018:**
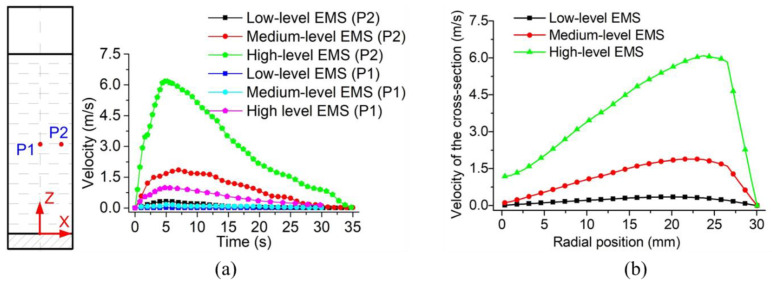
Variation of velocity (**a**) with time and (**b**) along the radius of the slurry.

**Figure 19 materials-13-05470-f019:**
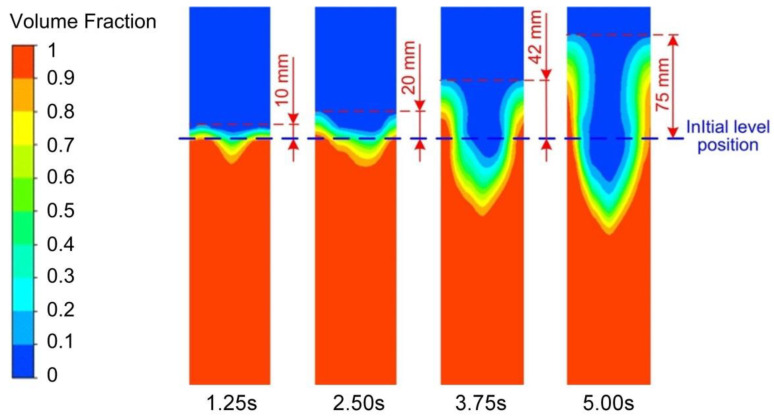
Vortex-shaped structure generation process under the high level of EMS.

**Figure 20 materials-13-05470-f020:**
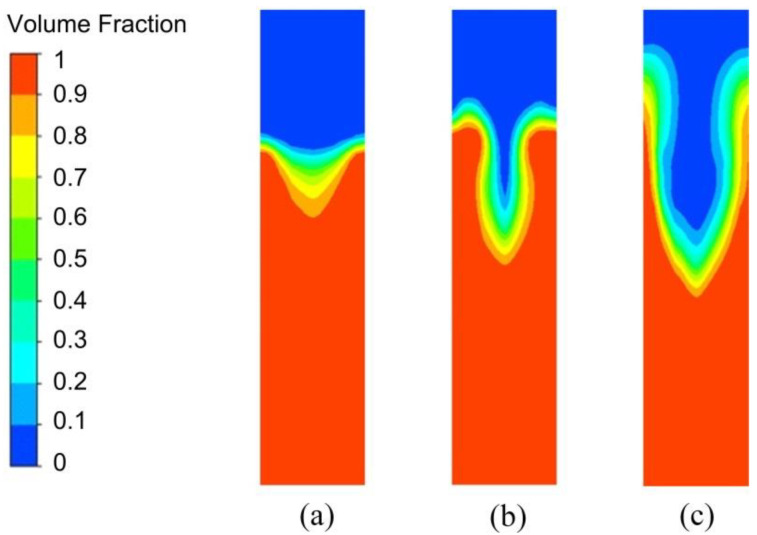
The vortex-shaped structure form in the vertical section under the (**a**) low, (**b**) medium, and (**c**) high levels of EMS.

**Figure 21 materials-13-05470-f021:**
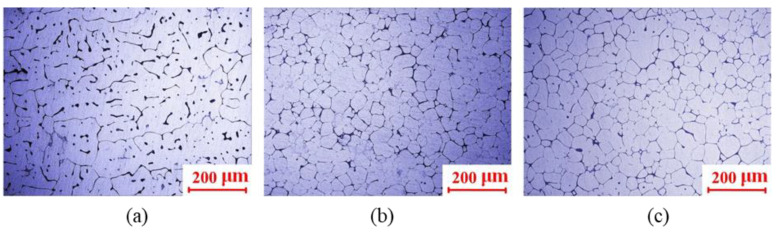
The microstructure of 2A50 alloy (**a**) prepared without EMS, (**b**) the central, and (**c**) edge samples of semi-solid 2A50 alloy prepared by EMS with the medium-level EMS.

**Figure 22 materials-13-05470-f022:**
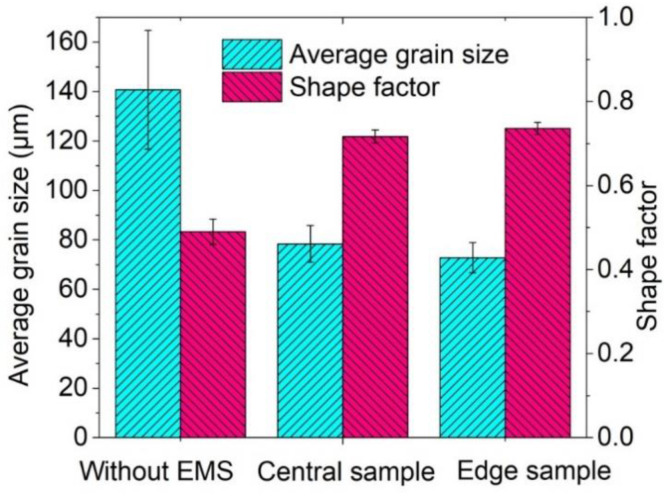
Average grain size and shape factor of the microstructure for the billet prepared without EMS and with the medium-level EMS.

**Table 1 materials-13-05470-t001:** Dimensions of the key parts of the electromagnetic stirrer and the aluminium alloy blank.

Item	External Diameter (mm)	Inner Diameter (mm)	Height (mm)
Yoke	328	182	180
Corundum tube	80	62	200
Stainless steel crucible	62	60	220
Aluminium alloy blank	60	-	180

**Table 2 materials-13-05470-t002:** Thermophysical parameters of materials in the numerical simulation.

Simulation Category	Parameter	Value
Electromagnetic field	Relative permeability of three-phase two-pole electromagnetic stirrer coil, $\varphi_{c}$	0.999991
	Resistivity of three-phase two-pole electromagnetic stirrer coil, $\sigma_{c}$ (Ω·m)	1.5 × 10^−7^
	Relative permeability of silicon steel yoke of three-phase two-pole electromagnetic stirrer, $\varphi_{s}$	2000
	Resistivity of silicon steel yoke of three-phase two-pole electromagnetic stirrer, $\sigma_{s}$ (Ω·m)	8.0 × 10^−6^
	Relative permeability of aluminium alloy paste, $\varphi_{a}$	1
	Resistivity of aluminium alloy paste, $\sigma_{s}$ (Ω·m)	2.1 × 10^−7^
	Relative permeability of stainless steel mixing drum and other parts, $\varphi_{o}$	1
	Resistivity of other parts such as stainless steel mixing drum, $\sigma_{o}$ (Ω·m)	9.1 × 10^−7^
Flow and temperature fields	Density of aluminium alloy paste, $\rho_{a}$ (kg/m^3^)	2405
	Thermal conductivity of aluminium alloy paste, ${Κ}_{a}$ (W/m·K)	204
	Specific heat capacity of aluminium alloy paste, $c_{a}$ (J/kg·K)	900
	Thermal conductivity of stainless steel mixing drum, ${Κ}_{s}$ (W/m·K)	26
	Specific heat capacity of stainless steel mixing drum, $c_{s}$ (J/kg·K)	500
	Aluminium alloy slurry-stainless steel mixing drum interface heat transfer coefficient, $h_{1}$ (W/m^2^·K)	1000
	Stainless steel mixing drum-air interface heat transfer coefficient, $h_{2}$ (W/m^2^·K)	10

**Table 3 materials-13-05470-t003:** Specifications of the intermediate frequency induction furnace.

Item	Specifications
Intermediate frequency electricity input	Three-phase, 380 V, 50 Hz
Intermediate frequency electricity output	Three-phase, 2000–3000 Hz, 0–380 V, 0–60 A
Rated power	25 kW (adjustable)
Control mode	PID closed-loop control
Maximum temperature difference	±5 °C
Rated power of the cooling water pump	1.5 kW
Flow rate of the cooling water	12.5 m^3^/h
Water pump head	20 m
Heating time	≤300 s

**Table 4 materials-13-05470-t004:** Specifications of aluminium alloy 2A50 [5].

Ingredient	Element	Cu	Si	Mg	Mn	Zn	Ti	Ni	Fe	Al
Proportion (wt.%)	2.43	0.82	0.68	0.53	0.12	0.06	0.05	0.01	Bal.
Temperature	Liquidus (°C)	615		
Solidus (°C)	521

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
