# Peer review of "Numerical Simulation and Experimental Investigation of the Preparation of Aluminium Alloy 2A50 Semi-Solid Billet by Electromagnetic Stirring"

_materials, 2020, doi:10.3390/ma13235470_

Round 1

Reviewer 1 Report

In Table 1 it would be appropriate to add a source of information on the composition of the aluminium alloy 2A50.
I would recommend using the words: “liquidus” and “solidus” instead of “phase line” and “liquidus solid phase line” in Table 4.
Figure 8 a is not clear sufficiently as values of electromagnetic intensity, induced current density and Lorentz force are described are shown in very lower case. Moreover, it would be appropriate to add units of physical quantities (e.g. Lorentz force in kN/m3, etc.).

Author Response

The authors would like to take this opportunity to sincerely thank the reviewer for his/her valuable comments. We have studied all the comments carefully and have made corresponding corrections that we hope will meet with your approval. The detailed responses to the reviewer’s comments are provided below.

  1. In Table 1 it would be appropriate to add a source of information on the composition of the aluminium alloy 2A50.

Response: Thanks for the reviewer’s comment. The source information and the supporting reference of the aluminum alloy 2A50 have been added to the content. The revised and added content is shown as follows.

Commercial aluminium alloy 2A50 extruded bar supplied by Aluminum Corporation of China was selected as the test material in this study. Its composition is summarised in Table 4, which shows that the main alloying element is copper.

Table 4. Specifications of aluminium alloy 2A50 [25].

Ingredient

Element

Cu

Si

Mg

Mn

Zn

Ti

Ni

Fe

Al

Proportion (wt%)

2.43

0.82

0.68

0.53

0.12

0.06

0.05

0.01

Bal.

Temperature

Liquidus (˚C)

615

Solidus (˚C)

521

Please see lines 215-218 and Table 4 on page 8 in the revised manuscript.

  1. I would recommend using the words: “liquidus” and “solidus” instead of “phase line” and “liquidus solid phase line” in Table 4.

Response: Thanks for pointing this out. “liquidus phase line” and “solid phase line” have been changed as “liquidus” and “solidus” in Table 4. The revised table is shown as follows.

Table 4. Specifications of aluminium alloy 2A50 [25].

Ingredient

Element

Cu

Si

Mg

Mn

Zn

Ti

Ni

Fe

Al

Proportion (wt%)

2.43

0.82

0.68

0.53

0.12

0.06

0.05

0.01

Bal.

Temperature

Liquidus (˚C)

615

Solidus (˚C)

521

Please see Table 4 on page 8.

  1. Figure 8 a is not clear sufficiently as values of electromagnetic intensity, induced current density and Lorentz force are described are shown in very lower case. Moreover, it would be appropriate to add units of physical quantities (e.g. Lorentz force in kN/m3, etc.).

Response: Thanks for the reviewer’s comment. Figure 8 has been revised with a larger font of values. The units of physical quantities have also been added in Figure 8. Please see the revision of Figure 8 on pages 11 and 12.

Reviewer 2 Report

In this work authors simulated the effects of electromagnetic stirring applied to an aluminum alloy billet.

simulation were performed to asses the shape of the primary pipe (or vortex as was called in the paper) and most significant electrical measurements.

the material related part is small with just a very basic comparison of a traditional as-cast microstructure and the ones obtained after EMS. 

the main conclusion is that by applying EMS dendritic microstructure with abundant segregation is substituted with an equiaxed one with a more limited copper segregation at grain boundaries (GB).

I think the biggest novelty in this paper may be the application of simulation to the EMS process since, magnetic stirring to ingots is a well known application.

some parts need to be rewritten since they are not clear.

most importantly it seems like the author who described the microstructure wrote the part by his own without discussing with the other authors. in all the initial part the material description is absolutely approximate. so please try to make the two parts more balanced by introducing some details also in the first part of the paper.

here the minor changes which I suggest before publication:

L 40-42 not clear at all. it may be eliminated since it does not give any useful information.

L 42 segregation of which element? i think copper...so please refer to the proper chemical element or elements which is/are segregating

L 43 which kind of Heat treatment are you meaning?

L 63-65 lamellar structure is the eutectic phase? why to a finer solidification structure corresponds a coarser grains? what do you mean by solidification structures? 

L 72-74 why what you are describing happens? please give a more precise explanation; the sentence is way too generic.

L 99 the vortex is described for the first time without explaining to which part of the ingot you're referring. please note that vortex is NOT the correct way to mention the typical cone formed during ingot solidification

L 108-109 the sentence is not clear please re-write

L 236 you're not corroding the material! it is called chemical etching!

L 241 grains should me measured according to ASTM E112. for as-cast microstructure the secondary dendritic arm spacing should be also mentioned

figure 7 why 3 samples are shown? which differences are there among them? this was not mentioned. what the colours means here is this something similar to figure 20? if yes, consider using only one of the two figures for brevity 

L272 273 why 50 A 50 Hz was chosen?

L273 281 this part is extremely difficult to be read. consider revising it with an English native speaker in order to make it more fluent. 

L 273 maximum and minimum values of electromagnetic intensity are observed at the surface...it seams that only maximum are located there. also the description of induced current is unclear it seams like the flux is focused on the external surface rather than in the center. each picture of figure 8 contains two cylinders what are the differences between the two? please describe it by adding some descriptive labels on them

L 470 in fig 21a I see a dendritic microstructure so, how grains were measured? please comment on this.

Author Response

The authors would like to take this opportunity to sincerely thank the reviewer for his/her valuable comments. We have studied all the comments carefully and have made corresponding corrections that we hope will meet with your approval. The detailed responses to the reviewer’s comments are provided below.

In this work authors simulated the effects of electromagnetic stirring applied to an aluminum alloy billet.

simulation were performed to asses the shape of the primary pipe (or vortex as was called in the paper) and most significant electrical measurements.

the material related part is small with just a very basic comparison of a traditional as-cast microstructure and the ones obtained after EMS. 

the main conclusion is that by applying EMS dendritic microstructure with abundant segregation is substituted with an equiaxed one with a more limited copper segregation at grain boundaries (GB).

I think the biggest novelty in this paper may be the application of simulation to the EMS process since, magnetic stirring to ingots is a well known application.

  1. some parts need to be rewritten since they are not clear.

most importantly it seems like the author who described the microstructure wrote the part by his own without discussing with the other authors. in all the initial part the material description is absolutely approximate. so please try to make the two parts more balanced by introducing some details also in the first part of the paper.

Response: Thanks for the reviewer’s comment. The microstructure discussion part has been revised in the current version of the manuscript. The revised and added content is shown as follows.

Lines 490-499 on page 20: Solid grains with the average grain size significantly smaller than 100μm were observed, which were evenly distributed in the eutectic phase. Moreover, some nearly spherical grains were observed bonding to each other in the microstructure of the central sample shown in Figure 21(b) compared to the edge sample shown in Figure 21(b), which can be attributed to the weak effect of the EMS at the central slurry.

Lines 516-528 on page 21: The achievement of fine and spherical semi-solid microstructures through the EMS process can be explained as follows. When the slurry was poured into the stirring chamber, the edge and bottom of the slurry directly contacted the wall of the stirring chamber, which rapidly caused the supercooling of the slurry at these contacting positions. Consequently, a large number of crystal nuclei are formed on the inner wall and bottom surface of the stirring chamber. Under the interference of the electromagnetic stirring, the slurry repeatedly washed the inner wall and bottom face due to the high-speed rotating. This washing movement of slurry forced the crystal nuclei generated to fall off and enter back to the slurry. Meanwhile, the slurry at the core space of the stirring drum was forced to the inner wall of the stirring chamber. This forced flow of slurry helped to form a uniform distribution of crystal nuclei, which can effectively inhibit the evolution of crystal nuclei to dendrites. Furthermore, the forced convection movement of the slurry can increase the local shear rate inside the stirring drum, which caused the frequent collision, rub, and shearing of grains and consequently formed the fine spherical grains.

here the minor changes which I suggest before publication:

  1. L 40-42 not clear at all. it may be eliminated since it does not give any useful information.

Response: Thanks for pointing this out. Lines 40-42 have been revised as “Aluminium alloy 2A50 has also been used for the production of wires as it has excellent heat conduction and corrosion resistance.” Please see lines 40-41 on page 1.

  1. L 42 segregation of which element? i think copper...so please refer to the proper chemical element or elements which is/are segregating

Response: Thanks for the comment. The relevant sentence has been revised by adding the chemical elements. The revised content is shown as follows.

However, the intergranular segregation of Cu, Mn, and Fe is a severe issue of aluminium alloy 2A50.

Please see lines 43-44 on pages 1 in the revised manuscript.

  1. L 43 which kind of Heat treatment are you meaning?

Response: L 43 has been revised by adding the T6 heat treatment. The revised content is shown as follows.

Although the T6 heat treatment strengthening can increase the hardness of aluminium alloy 2A50, the alloy elements are severely precipitated from the inside of the grains, resulting in a decrease in mechanical properties such as elongation and tensile strength.

Please see line 44 on page 1 in the revised manuscript.

  1. L 63-65 lamellar structure is the eutectic phase? why to a finer solidification structure corresponds a coarser grains? what do you mean by solidification structures? 

Response: Thanks for the comments. The interpretation has been corrected in this sentence. The revised content is shown as follows.

the increase in the magnetic strength assisted the grain refinement and uniformity of solidification microstructures. This refinement of the solidification microstructure resulted in a distinct increase in the grain boundary.

Please see lines 65-67 on page 2 in the revised manuscript.

  1. L 72-74 why what you are describing happens? please give a more precise explanation; the sentence is way too generic.

Response: Thanks for pointing this out. The sentences mentioned have been revised to have a better explanation. The revised content is shown as follows.

Mapelli et al. [21] investigated the effects of the winding current of the EMS on the pattern of flow streamlines in the liquid pool. It was concluded that the circumpherential motion of the flow was changed into the radial movement by increasing the current intensity, and therefore, the velocity field of the liquid bulk can be controlled by adjusting the winding current values.

Please see lines 72-76 on page 2 in the revised manuscript.

  1. L 99 the vortex is described for the first time without explaining to which part of the ingot you're referring. please note that vortex is NOT the correct way to mention the typical cone formed during ingot solidification

Response: Thanks for the reviewer’s comment. We have revised “the vortex” as “the vortex shaped structure” in the current version of the manuscript. The vortex-shaped structure is generated at the top surface of the billet, which has also been added in the revised manuscript. Please see lines 102-103 and112-113 on page 3.

  1. L 108-109 the sentence is not clear please re-write

Response: Thanks for pointing this out. This sentence has been revised in the current manuscript. The revised content is shown as follows.

The semi-implicit method for pressure linked equations (SIMPLE) is used in the simulation because the interface between the slurry and the inner wall is in a non-sliding shear state.

Please see lines 123-125 on page 4 in the revised manuscript.

  1. L 236 you're not corroding the material! it is called chemical etching!

Response: Thanks for the comment. The “corrosion” has been corrected to “chemical etching” in the mentioned sentences and the relevant content.

Please see lines 254-255 on pages 9 and 10, and Figure 5 on page 9 in the revised manuscript.

  1. L 241 grains should me measured according to ASTM E112. for as-cast microstructure the secondary dendritic arm spacing should be also mentioned

Response: Thanks for the reviewer’s comment. ASTM E112 and the quantitative measurement method are both widely used to analyze the solid grains. In this work, the quantitative measurement method is applied for the calculation of equivalent diameter and the shape factor of the grains.

In order to have a better interpretation, the content has been added, which is shown as follows.

The average diameter is the average value of the equivalent diameter of all grains. The equivalent diameter is the diameter of a circle, of which the area is equal to the area of the grain.

Please see lines 262-264 on page 10 of the revised manuscript.

This paper aims to investigate the effect of EMS on the grain size and shape factor. Therefore, the secondary dendritic arm spacing has not been measured in this work. But it is worthy to study the evolution of the secondary dendritic arm spacing through the EMS process, which will be investigated in our future study.

  1. figure 7 why 3 samples are shown? which differences are there among them? this was not mentioned. what the colours means here is this something similar to figure 20? if yes, consider using only one of the two figures for brevity 

Response: Thanks for the reviewer’s comment. Figure 7(b) shows the comparison of the vortex-shaped structure for one sample between the experimental and the simulation results under one operating condition (30A, 30Hz). In order to clearly compare the shape characteristics, three pictures are presented to show the difference in the shape line, maximum diameter, and the depth of the vortex-shaped structure. Figure 20 is different from Figure 7. Figure 20 compares the vortex-shaped structure formed at 3 operating conditions, which are 10A&10Hz, 30A&30Hz, and 50A&50Hz. This figure aims to reveal the effect of the operating conditions on the vortex-shaped structure.

In order to have a better interpretation, the content has been revised, which is shown as follows.

The comparison of the vortex-shaped structure for one sample between the experimental and the simulation calculated results with the winding current of 30 A and the electrical frequency of 30 Hz is presented in Figure 7.

In order to clearly compare the shape characteristics, three pictures are presented to show the difference in the shape line, maximum diameter, and the depth of the vortex-shape structure.

Please see lines 274-276 and 279-281 on page 10.

  1. L272 273 why 50 A 50 Hz was chosen?

Response: Thanks for the reviewer’s comment. The operating condition of 50 A and 50 Hz were chosen for presenting the magnetohydrodynamics analysis as an example as this is the maximum winding current and electrical frequency studied in this paper. The explanation has been added in the revised manuscript. The revised content is shown as follows.

The operating conditions of 50 A and 50 Hz are chosen as an example to show the magnetohydrodynamics results as they are the maximum operating conditions in this study.

Please see 294-296 on page 11.

  1. L273 281 this part is extremely difficult to be read. consider revising it with an English native speaker in order to make it more fluent. 

Response: Thanks for the reviewer’s comment. This content has been revised for a better description with the assistance of an English native speaker. The revised content is shown as follows.

It can be seen from Figure 8(a) that the maximum and the minimum values of the electromagnetic intensity are all observed at the outside surface of the cylinder billet. The electromagnetic intensity at the top and bottom surfaces, as well as the outside cylindrical surface, were all higher than the interior electromagnetic intensity, which represents the surface and end effects. Figure 8(b) shows that the vectors of the induced current form closed loops at both the outside surface and interior region of the billet. The distribution pattern of the induced current density is similar to the electromagnetic intensity, which also shows the surface and end effects. It can be found in Figure 8(c) that the Lorentz force distribution presents more severe surface and end effects. The minimum value of the Lorentz force occupied the majority of the interior region of the billet while the maximum was observed at the outside surface of the slurry.

Please see lines 296-306 on page 11.

  1. L 273 maximum and minimum values of electromagnetic intensity are observed at the surface...it seams that only maximum are located there. also the description of induced current is unclear it seams like the flux is focused on the external surface rather than in the center. each picture of figure 8 contains two cylinders what are the differences between the two? please describe it by adding some descriptive labels on them

Response: Thanks for the reviewer’s comment. The two cylinders in Figure 8 shows the outside overview and the vertical section view of the slurry, respectively. Figure 8 has been revised by adding more descriptions to avoid any misunderstanding.

Please see Figure 8 on page 12 in the revised manuscript.

  1. L 470 in fig 21a I see a dendritic microstructure so, how grains were measured? please comment on this.

Response: In this work, the grain size is given in its average diameter. The average diameter is the average value of the equivalent diameter of all grains. The equivalent diameter is the diameter of a circle of which the area is equal to the area of the grain. This explanation has been added in the revised manuscript. Please see lines 262-264 on page 10.

Reviewer 3 Report

In general, the work is interesting and can make a significant contribution to the metallurgical processes of alloys with a homogeneous structure. However, there is one significant remark to the article. The authors call up experimental results, but do not comment on their connection with the calculations. Do the calculated models allow predicting the final alloy structure? If so, how and what is the model-experiment relationship. In my opinion, the authors take a more responsible approach to the interpretation of experimental data, in which there is still no discussion.

Author Response

The authors would like to take this opportunity to sincerely thank the reviewer for his/her valuable comments. We have studied all the comments carefully and have made corresponding corrections that we hope will meet with your approval. The detailed responses to the reviewer’s comments are provided below.

  1. In general, the work is interesting and can make a significant contribution to the metallurgical processes of alloys with a homogeneous structure. However, there is one significant remark to the article. The authors call up experimental results, but do not comment on their connection with the calculations. Do the calculated models allow predicting the final alloy structure? If so, how and what is the model-experiment relationship. In my opinion, the authors take a more responsible approach to the interpretation of experimental data, in which there is still no discussion.

Response: Thanks for the reviewer’s comment.

We agree with the reviewer that the calculated models cannot be used for predicting the final alloy structure. However, the simulation can reveal the heat and mass transfer of the slurry under the influence of EMS. This thermal process and liquid flow in the slurry pool are vital to determine the billet geometry and solidification microstructure. Therefore, the integrated and quantitative description of the slurry behavior is necessary to provide a deep insight into the physical mechanism of EMS. However, experimental investigation on the thermos-physical information in the slurry pool during the EMS is an extremely difficult task due to the non-transparent feature of the semi-solid slurry. Thus, the simulation in this study is provided a deep understanding of the thermos-physical characteristics under different operating conditions while the experimental results show the final microstructure results under these operating conditions. This explanation has been added in the revised manuscript. Please see lines 84-93 on page 2.  

More discussion on the experimental results has been added in the revised manuscript. The revised content is shown as follows.

The achievement of fine and spherical semi-solid microstructures through the EMS process can be explained as follows. When the slurry was poured into the stirring chamber, the edge and bottom of the slurry directly contacted the wall of the stirring chamber, which rapidly caused the supercooling of the slurry at these contacting positions. Consequently, a large number of crystal nuclei are formed on the inner wall and bottom surface of the stirring chamber. Under the interference of the electromagnetic stirring, the slurry repeatedly washed the inner wall and bottom face due to the high-speed rotating. This washing movement of slurry forced the crystal nuclei generated to fall off and enter back to the slurry. Meanwhile, the slurry at the core space of the stirring drum was forced to the inner wall of the stirring chamber. This forced flow of slurry helped to form a uniform distribution of crystal nuclei, which can effectively inhibit the evolution of crystal nuclei to dendrites. Furthermore, the forced convection movement of the slurry can increase the local shear rate inside the stirring drum, which caused the frequent collision, rub, and shearing of grains and consequently formed the fine spherical grains.

Please see lines 516-528 on page 21.
